# Asymmetric Tilt-Induced Quantum Beating of Conductance Oscillation in Magnetically Modulated Dirac Matter Systems

**DOI:** 10.3390/nano14090811

**Published:** 2024-05-06

**Authors:** Nawapan Sukprasert, Patchara Rakrong, Chaiyawan Saipaopan, Wachiraporn Choopan, Watchara Liewrian

**Affiliations:** 1Secondary Science Division, Saint Gabriel’s College, Bangkok 10300, Thailand; 2Theoretical and Computational Physics Group (TCP), Science Laboratory Building, Department of Physics, Faculty of Science, King Mongkut’s University of Technology Thonburi, Bangkok 10140, Thailand; 3Theoretical and Computational Science Center (TaCS), Science Laboratory Building, Faculty of Science, King Mongkut’s University of Technology Thonburi (KMUTT), Bangkok 10140, Thailand; 4Demonstration School, Bansomdejchaopraya Rajabhat University, Bangkok 10600, Thailand; 5Department of Biomedical Engineering, College of Health Science, Christian University of Thailand, Nakhonpathom 73000, Thailand

**Keywords:** Dirac materials, graphene, quantum beat, spin polarization, tilted Dirac cone

## Abstract

Herein, we investigate the effect of tilt mismatch on the quantum oscillations of spin transport properties in two-dimensional asymmetrically tilted Dirac cone systems. This study involves the examination of conductance oscillation in two distinct junction types: transverse- and longitudinal-tilted Dirac cones (TTDCs and LTDCs). Our findings reveal an unusual quantum oscillation of spin-polarized conductance within the TTDC system, characterized by two distinct anomaly patterns within a single period, labeled as the linear conductance phase and the oscillatory conductance phase. Interestingly, these phases emerge in association with tilt-induced orbital pseudo-magnetization and exchange interaction. Our study also demonstrates that the structure of the LTDC can modify the frequency of spin conductance oscillation, and the asymmetric effect within this structure results in a quantum beating pattern in oscillatory spin conductance. We note that an enhancement in the asymmetric longitudinal tilt velocity ratio within the structure correspondingly amplifies the beating frequency. Our research potentially contributes valuable insights for detecting the asymmetry of tilted Dirac fermions in type-I Dirac semimetal-based spintronics and quantum devices.

## 1. Introduction

Graphene, the first two-dimensional carbon to be realized [1,2,3,4], has sparked great interest in condensed matter physics, particularly in the study of charge transport phenomena in two-dimensional (2D) systems. This interest has resulted in the identification of a novel class of materials called Dirac materials [5,6]. These materials exhibit unique transport properties, such as the chiral nature of the electronic wave function and a linear band structure around the Dirac point. At this point, the behavior of charge carriers exhibits characteristics analogous to relativistic massless Dirac fermions, a behavior governed by the massless Dirac equation. One of the intriguing features of Dirac materials is the Dirac cone, a term referring to the linear energy dispersion relation around the Dirac point. In graphene, this cone is isotropic, meaning the energy dispersion is the same in all directions. However, the Dirac cone is sensitive to changes in the local crystal structure and symmetry [7,8], leading to potential deviations from Lorentz invariance due to strong anisotropy in the band structure. Recent research reveals unprecedented control over Dirac cones in graphene and other emerging 2D materials. DFT calculations reveal that the hydrogenation of graphene leads to the emergence of a tilted and anisotropic Dirac cone [9]. Furthermore, quinoid-type deformations in graphene lead to a linear relationship between strain and Dirac cone tilt [10]. This tunability extends beyond graphene; tilted Dirac cones arise in materials like α-(BEDT-TTF)_2_I_3_ under uniaxial pressure [11] and 8-Pmmn borophene, which intrinsically exhibits anisotropic, tilted cones [12]. Engineering these properties has profound implications: coupled periodic scalar and vector potentials enable tunable tilted anisotropy in graphene-based structures [13]. Even atomic-level modifications within 8-Pmmn borophene offer precise control over the Dirac cone’s tilt [14]. Notably, tilted Dirac cones in photonic systems lead to entirely new phenomena. Using a Lieb–kagome lattice framework, Hanafi et al. demonstrated the manipulation of Dirac cone tilt in photonic structures, unveiling asymmetric conical diffraction and showcasing control confirmed both theoretically and experimentally [15]. Photonic systems provide a rich platform for exploring Dirac cone physics. Tilted Dirac cones in artificial photonic systems are designed to mimic the behavior of electrons in graphene. These can be controlled by manipulating the interplay between flat bands and Dirac cones, as demonstrated in photonic orbital graphene [16]. This control suggests numerous intriguing research avenues, from fundamental explorations of exotic Dirac physics to the design of functional devices exploiting engineered tilt characteristics.

Tilting Dirac cones disrupts their perfect conical symmetry, introducing anisotropy. This anisotropy significantly impacts the material’s transport properties, such as conductivity [17], magnetoresistance [18], and the Hall effect [19]. Understanding how the tilt influences these properties is crucial not only for characterizing these materials but also for manipulating their behavior for potential device applications. Notably, this anisotropy enables the perfect transmission of charge carriers at non-zero angles of incidence, the hallmark of anomalous Klein tunneling [20]. The tilt-induced separation of charge carriers by chirality in Dirac/Weyl systems enables the design of devices based on valley filtering and beam splitting [21]. Tilting the Dirac cone in graphene-based Josephson junctions leads to electrically charged Andreev modes [22]. These modes, involving cooperative electron–hole reflection, offer exciting potential for novel quantum devices and fundamental physics exploration. Unexpectedly, the Josephson current scales as N² (where N represents transmission channels), rather than linearly. This reflects the unique behavior of charged Andreev modes in tilted Dirac materials, where the tilt acts like a strong gravitational field, promoting the generation and repeated reflection of electron–hole pairs. Furthermore, the tilted Dirac cone leaves measurable signatures, offering ways to characterize its properties and potentially manipulate a material’s electronic behavior. For example, in α-(BEDT-TTF)_2_I_3_, analyzing the interlayer magnetoresistance can reveal the strength and direction of the Dirac cone tilt. [18]. Additionally, it has been reported that the Fano factor, a measure of the fluctuations in a system, is vulnerable to the tilt of the Dirac cone [23]. This suggests that the Fano factor could be used as a tool to validate the tilted Dirac cone characteristics of materials.

A key discovery in this field is the oscillatory behavior of spin conductance in ferromagnetic-gate graphene structures [24]. This spin conductance can be tuned through the gate potential. Further research demonstrated that the exchange interaction appears to generate quantum modulation [25]. Additionally, the existence of quantum beats, a result of interference between oscillatory spin conductance components, arises from the ferromagnetic graphene barrier [26]. This demonstrates the profound impact of the exchange field on the transport properties of graphene. Interestingly, it was observed that the amplitudes of quantum oscillation do not diminish with thicker barriers, a finding relevant to graphene-based device design [25]. The potential for controllable spin transport in double ferromagnetic-gate graphene has been theoretically demonstrated [27], suggesting that ferromagnetic gates could be used to engineer graphene’s transport properties. Finally, recent work indicates a strong influence of tilted Dirac cones on the frequency of spin conductance oscillation in single tilted Dirac cones coupled with ferromagnetic graphene [28].

The ability to manipulate spin conductance and observe the resulting quantum oscillation phenomena opens up possibilities for understanding and controlling spin transport properties in graphene, as demonstrated by phenomena like the Shubnikov–de Haas (SdH) effect. In normal metals, the SdH oscillations occur when a magnetic field is applied [29]. Electrons in the conduction band exhibit oscillatory behavior as Landau levels shift and periodically intersect the Fermi energy. This results in fluctuations in the electron population at the Fermi surface. The result reveals the quantum oscillations in the metal. Yampol’skii et al. [30] showed that the massless Dirac fermions in graphene can be shown quantum oscillations by driving an external electric field, like a magnetic field SdH-like oscillation. This oscillation arises due to the periodic movement of the localized levels passing over the Fermi energy, leading to a change in the density of localized states in the electrostatic potential barrier. The asymmetry of double potential barriers in graphene heterostructures introduces a difference in the frequency or phase of the quantum oscillations associated with each barrier. This difference gives rise to the phenomenon of quantum beats. Liu et al. [31] studied SdH-like oscillations of conductance in graphene-based structures with double Fermi velocity barriers. They discovered that the quantum beat of charge conductance directly arises from the asymmetry of the double velocity modulation. The presence of quantum beats offers a potentially sensitive method for detecting and characterizing subtle asymmetries within heterostructures. The introduction of an exchange field further modifies the quantum beat phenomenon, influencing spin-polarized transport and causing a shift in the spin-dependent quantum beats. Saipaopan et al. [32] observed similar SdH-like oscillations in graphene-based structures with both double Fermi velocity modulation and an exchange field. Significantly, they discovered the existence of quantum beats in spin-resolved conductance. This beating behavior arises from the superposition of the oscillating conductances within the asymmetric double Fermi velocity barriers. Crucially, the presence of the exchange field has a remarkable effect: it causes the spin-resolved quantum beats to shift in the opposite direction. The discovery of exchange field-induced shifts in spin-resolved quantum beats underscores the potential for the fine control of spin transport properties in graphene heterostructures. 

In this study, we investigate the spin transport properties of Dirac electrons in asymmetric tilted Dirac cone heterostructures under the influence of an exchange field. Our primary focus is to analyze how the tilt mismatch between two Dirac cones affects the quantum oscillations observed in spin transport properties within these two-dimensional systems. This analysis will involve examining conductance oscillations in both transverse- and longitudinal-tilted Dirac cone junctions. Our model incorporates a unique feature: tilted Dirac cones exhibiting both transverse and longitudinal tilts. In a transverse-tilted Dirac cone (TTDC), the tilt is perpendicular to the electron’s direction of motion. Conversely, in a longitudinal-tilted Dirac cone (LTDC), the tilt is parallel to the electron’s motion. These distinct tilting directions can significantly influence the spin transport properties within the heterostructure. Next, we aim to explore the voltage-driven quantum oscillation of spin-resolved conductance within the TTDC system. We are particularly interested in identifying any distinct anomaly patterns (e.g., sudden shifts, unexpected peaks) within a single oscillation period that may arise due to the interplay between tilt-induced orbital pseudo-magnetization and exchange interaction. Finally, we plan to investigate how the specific structure of the LTDC cone can modify the frequency of spin conductance oscillation. We are also interested in understanding how the asymmetric effect within this structure might result in a quantum beating pattern (characterized by multiple overlapping frequencies) in oscillatory spin conductance. Building upon the investigation of spin transport properties, our model of a tilted Dirac cone draws inspiration from experimentally realized systems like graphene–borophene heterostructures [33]. In these structures, lattice mismatch induces tilt and anisotropy in the Dirac cone. Additionally, normal-buckled graphene superlattices [34] demonstrate how controlled deformations can modulate the Dirac cone shape. Analyzing quantum beats could provide valuable insights into the symmetry of tilted heterostructures, aiding in the optimization and control of their properties for anisotropic device applications.

## 2. Materials and Methods

As illustrated in Figure 1, our model system consists of massless Dirac fermions traveling through a junction with alternating non-tilted (ND) and tilted Dirac cone (TDC) regions: ND/TDC1/ND/TDC2/ND. Notably, both TDC regions exhibit both transverse and longitudinal tilts. A magnetic insulator coating induces exchange fields ($J_{1}$ and $J_{2}$) in TDC1 and TDC2, respectively. These exchange fields can be further tuned by applying a gate potential ($U_{1}$ for TDC1 and $U_{2}$ for TDC2) via a metallic gate deposited on the insulator. The widths of TDC1 and TDC2 are $L_{1}$ and $L_{2}$, respectively, and are separated by a distance *D*. It is important to note that this model focuses on a scenario where the tilted Dirac cone regions exhibit both transverse and longitudinal tilts. If we consider a case where the tilts are purely transverse (TTDC) or purely longitudinal (LTDC), the corresponding junction configuration would be ND/TTDC1/ND/TTDC2/ND or ND/LTDC1/ND/LTDC2/ND, as depicted in Figure 1a,b, respectively. These variations would allow us to isolate the effects of each tilt direction on the spin transport properties. 

While the exchange interaction with magnetic ions in magnetic insulators has been recognized to influence the electronic structure of Dirac materials, we assume in this study that such an interaction does not significantly affect the electronic structure of the tilted Dirac cone material. This assumption is based on the proposition that the exchange interaction arises from the orbital overlap between the tilted Dirac material and the magnetic ions in the unoccupied shells of the magnetic insulator. Notably, the direction of the exchange field is consistently identified as perpendicular to the graphene plane, as supported by multiple studies [35,36,37,38,39,40].

Our model exploits the analogy between low-energy quasiparticles in Dirac materials and relativistic quantum particles. This allows us to simplify the model by assuming the propagation of massless Dirac electrons in the ballistic transport regime, where spin-flip scattering is negligible [27,41]. Additionally, intervalley scattering between K and K′ valleys within the Brillouin zone is considered insignificant [42,43]. Based on these assumptions, we can derive a general Dirac Hamiltonian that incorporates both transverse and longitudinal tilts. This Hamiltonian effectively captures the anisotropic nature of the tilted Dirac cones within our model system.
(1)$$\hat{H}=\sum_{i,j}\left(\hbar v_{ij}\kappa_{i}\sigma_{j}+\hbar \left(\rho_{i1,2}\kappa_{i}-V_{\tau }\left(x\right)\right)\sigma_{0}\right)$$
where the index $i,j\in \left\{x,y\right\}$. The potential can be defined as $V_{\tau }\left(x\right)=E_{F}+U_{1,2}+\Lambda_{\tau }J_{1,2}$ for TDC1, TDC2 regions and $V_{\tau }\left(x\right)=E_{F}$ for ND regions. Spin-up and spin-down electrons are represented as $\Lambda_{↑,↓}=+1,-1$. In tilted Dirac cones with both transverse and longitudinal tilts, the wave vector, denoted by $\kappa_{i}$, becomes directionally dependent due to the altered dispersion relation. This alteration transforms the originally circular Fermi surface (characteristic of non-tilted cones) into an elliptical Fermi surface, as illustrated in Figure 2, while $\sigma_{x,y}$ are the Pauli matrices and $\sigma_{0}$ is the 2 × 2 unit matrix. All anisotropies can be described using the upper diagonal of the anisotropy matrix $v_{ij}$. The tilt term $\rho_{i}\kappa_{i}$ arises when the tilted Dirac cone depends on the tilt velocity $\rho_{x1,2}$ and $\rho_{y1,2}$, representing LTDC1(2) and TTDC1(2), respectively. In this study, we focus solely on the tilted Dirac cone. The anisotropy matrix is fixed at $v_{ij}=v_{F}\delta_{ij}$, where $v_{F}\approx 10^{6}$ m/s. In the context of the scattering problem, the propagation of massless Dirac fermions is across the asymmetric tilted Dirac cone. The tilt velocity in each region can be written as follows:(2)$$\rho_{i1,2}=\left\{\begin{array}{l} 0,\;x<-L_{1}-D/2\\ \rho_{i1},\;-L_{1}-D/2\leq x\leq -D/2\\ 0,\;-D/2\leq x\leq D/2\\ \rho_{i2},\;D/2\leq x\leq L_{2}+D/2\\ 0,\;x>L_{2}+D/2 \end{array}\right.$$

The wave function for a non-tilted Dirac cone in each *ND* region described as
(3)$$\begin{array}{l} \psi_{I}\left(x<-L_{1}-D/2\right)=\left(\begin{array}{c} 1\\ e^{i\phi } \end{array}\right)e^{ik_{x}x}+r_{\tau }\left(\begin{array}{c} 1\\ -e^{i\phi } \end{array}\right)e^{-ik_{x}x}\\ \psi_{III}\left(-D/2\leq x\leq D/2\right)=c_{\tau }\left(\begin{array}{c} 1\\ e^{i\phi } \end{array}\right)e^{ik_{x}x}+d_{\tau }\left(\begin{array}{c} 1\\ -e^{i\phi } \end{array}\right)e^{-ik_{x}x}\\ \psi_{V}\left(x>L_{2}+D/2\right)=t_{\tau }\left(\begin{array}{c} 1\\ e^{i\phi } \end{array}\right)e^{ik_{x}x} \end{array}$$

The wave vector component of particles outside barrier is denoted as $k_{x}=\left(E_{F}/\hbar v_{F}\right)\cos\phi =k_{F}\cos\phi$, $k_{y}=\left(E_{F}/\hbar v_{F}\right)\sin\phi =k_{F}\sin\phi$. ϕ denotes the incident angle with respect to the barrier. Next, the wave function in the TDC1, TDC2 regions is written as
(4)$$\begin{array}{l} \psi_{II}^{\tau }\left(-L_{1}-D/2\leq x\leq -D/2\right)=a_{\tau }\left(\begin{array}{c} 1\\ \eta_{1} \end{array}\right)e^{iq_{x}^{\tau ,+}x}+b_{\tau }\left(\begin{array}{c} 1\\ \eta_{2} \end{array}\right)e^{-iq_{x}^{\tau ,-}x}\\ \psi_{IV}^{\tau }\left(D/2\leq x\leq L_{2}+D/2\right)=f_{\tau }\left(\begin{array}{c} 1\\ \eta_{3} \end{array}\right)e^{ip_{x}^{\tau ,+}x}+g_{\tau }\left(\begin{array}{c} 1\\ \eta_{4} \end{array}\right)e^{-ip_{x}^{\tau ,-}x} \end{array}$$
where $\eta_{1,2}=-\frac{q_{x}^{\tau ,\pm }\pm iq_{y}}{\mp \sqrt{{\left(q_{x}^{\tau ,\pm }\right)}^{2}+{q_{y}}^{2}}}$ and $\eta_{3,4}=-\frac{p_{x}^{\tau ,\pm }\pm ip_{y}}{\mp \sqrt{{\left(p_{x}^{\tau ,\pm }\right)}^{2}+{p_{y}}^{2}}}$. The wave vector components of particles inside TDC1 and TDC2 are $q_{x}^{\tau ,\pm }=q\cos\theta_{1}^{\tau ,\pm }$, $q_{y}=q\sin\theta_{1}^{\tau ,\pm }$, and $p_{x}^{\tau ,\pm }=p\cos\theta_{2}^{\tau ,\pm }$, $p_{y}=p\sin\theta_{2}^{\tau ,\pm }$. The refractive angle for electron propagation in TDC1 and TDC2 can be written as $\theta_{1}^{\tau ,\pm }=\arctan\left(q_{y}/q_{x}^{\tau ,\pm }\right)$, $\theta_{2}^{\tau ,\pm }=\arctan\left(p_{y}/p_{x}^{\tau ,\pm }\right)$. We can find the wave vector component inside barriers by considering the corresponding eigenenergy of the tilting Dirac cone in the Dirac material under the exchange field. Energy dispersion from Equation (1) can be written as
(5)$$E_{\lambda }\left(\kappa \right)=-V_{\tau }\left(x\right)+\hbar \rho_{x1,2}\kappa_{x}+\hbar \rho_{y1,2}\kappa_{y}+\lambda \hbar \sqrt{{\left(v_{F}\kappa_{x}\right)}^{2}+{\left(v_{F}\kappa_{y}\right)}^{2}}$$
where λ is the band index. The tilting of the Dirac cone can be represented by the vector ${\vec{\rho }}_{1,2}=\left(\rho_{x1,2},\rho_{y1,2}\right)$ that corresponds to the tilted magnitude $\rho_{1,2}=\sqrt{{\rho_{x1,2}}^{2}+{\rho_{y1,2}}^{2}}$ along the angle $\gamma =\arctan\left(\rho_{y1,2}/\rho_{x1,2}\right)$ with respect to the $\kappa_{x}$ axis. To effectively investigate the quantum beating pattern, the transverse-tilted velocities ($\rho_{y1}$ and $\rho_{y2}$) must be zero. While the longitudinal-tilted velocities ($\rho_{x1}$ and $\rho_{x2}$) should ideally range between $0.2v_{F}$ and $0.4v_{F}$ for enhanced pattern visibility, the pattern can still be observed within a broader range. To observe the wave vector of the tilted Dirac cone under the zero-bias condition, it should be ensured that $E_{\lambda }$ = 0 [24]. Equation (5) is written as $\frac{{\left(\kappa_{x\prime }+k_{c}\right)}^{2}}{{k_{a}}^{2}}+\frac{{\kappa_{y\prime }}^{2}}{{k_{b}}^{2}}=1$ with the components $k_{a}=\frac{\left(E_{F}+U_{1,2}+\Lambda_{\tau }J_{1,2}\right)}{\lambda \hbar v_{F}\left(1-\beta^{2}\right)}$, $k_{b}=\frac{\left(E_{F}+U_{1,2}+\Lambda_{\tau }J_{1,2}\right)}{\lambda \hbar v_{F}\sqrt{\left(1-\left(\alpha^{2}/\beta^{2}\right)\right)\left(1-\beta^{2}\right)}}$, and $k_{c}=\frac{\left(E_{F}+U_{1,2}+\Lambda_{\tau }J_{1,2}\right)\beta }{\lambda \hbar v_{F}\left(1-\beta^{2}\right)}$. The parameters are given by $\alpha =\frac{2\rho_{x1,2}\rho_{y1,2}}{\lambda^{2}{v_{F}}^{2}}$, $\beta =\frac{\rho_{1,2}}{\lambda v_{F}}$. The angular-dependent wave vector can be represented in the following form:(6)$$\kappa =\frac{k_{a}\left(1-{w_{0}}^{2}\right)}{1+w_{0}\cos\left(\phi -\gamma \right)}=\frac{\left(E_{F}+U_{1,2}+\Lambda_{\tau }J_{1,2}\right)}{\hbar v_{F}\left(\lambda +\left(\rho_{x1,2}/v_{F}\right)\cos\phi +\left(\rho_{y1,2}/v_{F}\right)\sin\phi \right)}$$

For the Fermi surface, $w_{0}=k_{c}/k_{a}=\beta$ is the eccentricity. λ indicates that the direction of the conduction band and the valance band are always tilted in opposite directions. The wave vector in Equation (6) reveals that the angular-dependent properties disappear when the Dirac cone is non-tilted [44]. Equation (6) also depicts how the angular-dependent wave vector outside the barrier depends on the incident angle. The expression for the angular-dependent wave vector inside the barrier in a function of the refractive angle is as follows:(7)$$\kappa =\frac{\left(E_{F}+U_{1,2}+\Lambda_{\tau }J_{1,2}\right)}{\hbar v_{F}\left(\lambda +\left(\rho_{x1,2}/v_{F}\right)\cos\theta +\left(\rho_{y1,2}/v_{F}\right)\sin\theta \right)}$$

The refractive angle $\theta =\sin^{-1}\left(\left(k_{F}/\kappa \right)\sin\phi \right)$ is determined by considering the translational symmetry in the y-direction, where the wave vector parallel to the y-axis remains conserved. For simplification, both barriers have the same exchange field tuning with the same gate voltage $U_{1}=U_{2}=U$ and $J_{1}=J_{2}=J$. After substituting θ into Equation (7), we obtain the x-component of the wave vector for the particle inside TDC1 and TDC2 as
(8)$$\begin{array}{l} q_{x}^{\tau ,\pm }=\frac{1}{L_{1}}\left(A_{1}\mp B_{1}\right)=\frac{1}{L_{1}}\left(\frac{\xi_{x1}\omega_{1}}{1-{\xi_{x1}}^{2}}\mp \frac{\sqrt{{\omega_{1}}^{2}+{\left(k_{F}L_{1}\sin\phi \right)}^{2}\left(1-{\xi_{x1}}^{2}\right)}}{1-{\xi_{x1}}^{2}}\right)\\ p_{x}^{\tau ,\pm }=\frac{1}{L_{2}}\left(A_{2}\mp B_{2}\right)=\frac{1}{L_{2}}\left(\frac{\xi_{x2}\omega_{2}}{1-{\xi_{x2}}^{2}}\mp \frac{\sqrt{{\omega_{2}}^{2}+{\left(k_{F}L_{2}\sin\phi \right)}^{2}\left(1-{\xi_{x2}}^{2}\right)}}{1-{\xi_{x2}}^{2}}\right) \end{array}$$
where $\omega_{1,2}=k_{F}L_{1,2}+\chi_{1,2}+\Lambda_{\tau }\mu_{1,2}-\xi_{y1,2}k_{F}L_{1,2}\sin\phi$. We defined $k_{F}L_{1,2}=\frac{E_{F}L_{1,2}}{\hbar v_{F}}$, $\chi_{1,2}=\frac{UL_{1,2}}{\hbar v_{F}}=\frac{k_{F}L_{1,2}U}{E_{F}}$, and $\mu_{1,2}=\frac{JL_{1,2}}{\hbar v_{F}}=\frac{k_{F}L_{1,2}J}{E_{F}}$. The parameters $\xi_{x1,2}=\rho_{x1,2}/v_{F}$ and $\xi_{y1,2}=\rho_{y1,2}/v_{F}$ are the longitudinal tilt velocity ratio and transverse tilt velocity ratio in the TDC1 and TDC2 regions, respectively. The width of both barriers is equal, $L_{1}=L_{2}=L$. The spin and tilt dependence of the transmission coefficient $t_{\tau ,\xi }$ can be determined by considering the wave function’s continuity at the boundaries located at *x* = −*L* − *D*/2, *x* = −*D*/2, *x* = *D*/2, and *x* = *L* + *D*/2. In the limit of $E_{F}\ll \left|U+\Lambda_{\tau }J\right|$ and $\theta_{1,2}^{\tau ,\pm }$ = 0, one can observe the oscillatory behavior of conductance easily. The transmission coefficient is as follows:(9)$$t_{\tau ,\xi }=\frac{e^{i\left(A_{1}+A_{2}\right)}e^{-ik_{F}(2L+D)\cos\phi }\cos^{2}\phi }{e^{ik_{F}D\cos\phi }\sin B_{1}\sin B_{2}\sin^{2}\phi +e^{-ik_{F}D\cos\phi }\left(\cos B_{1}\cos\phi +i\sin B_{1}\right)\left(\cos B_{2}\cos\phi +i\sin B_{2}\right)}$$

**Figure 2 nanomaterials-14-00811-f002:**
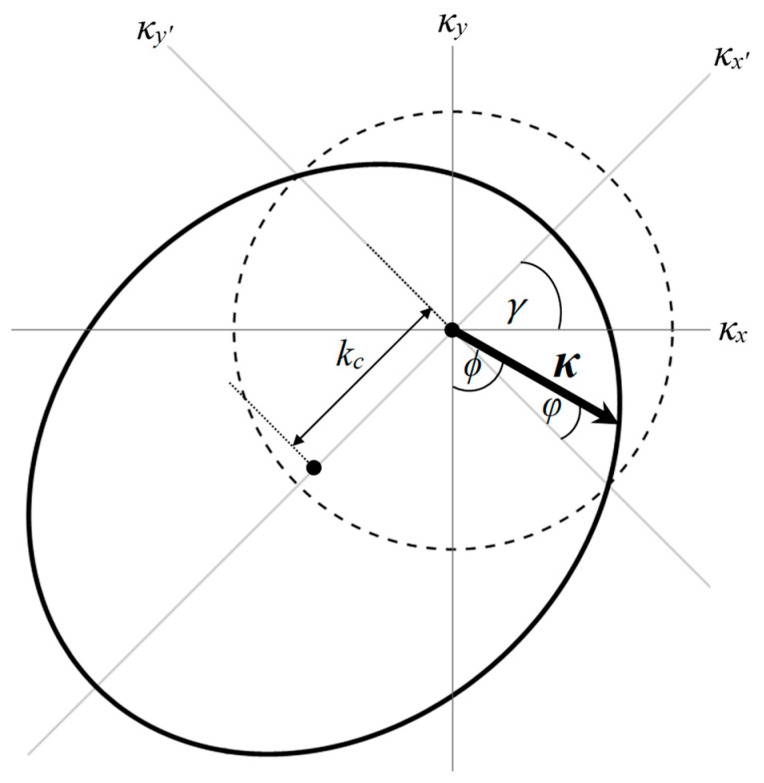
Comparison of Fermi surface cross-sections for non-tilted and tilted Dirac cones. The dashed line represents the circular cross-section of a non-tilted Dirac cone, while the solid line depicts the elliptical cross-section resulting from the presence of tilts in the Dirac cone. This transformation arises due to the altered dispersion relation caused by the presence of both transverse and longitudinal tilts in the tilted cone. As a result, the wave vector (denoted by $\kappa_{i}$) becomes directionally dependent, influencing the overall shape of the Fermi surface. The tilted velocity represented by the dashed line is zero and the solid line is ${\vec{\rho }}_{1,2}=\left(0.4v_{F},0.4v_{F}\right)$.

Finally, the spin-resolved conductance $G_{\tau ,\xi }$ in terms of the transmission probability $T_{\tau ,\xi }\left(\phi \right)$ can be written as [24]
(10)$$G_{\tau ,\xi }=G_{0}\int_{-\pi /2}^{\pi /2}T_{\tau ,\xi }\left(\phi \right)\cos\phi d\phi =G_{0}{\int_{-\pi /2}^{\pi /2}\left|t_{\tau ,\xi }\left(\phi \right)\right|}^{2}\cos\phi d\phi$$
where $G_{0}=2e^{2}E_{F}L_{y}/\pi \hbar$ with the width of structure $L_{y}$. The important parameters are the charge conductance $G_{C}=G_{↑}+G_{↓}$ and the spin conductance $G_{s}=G_{↑}-G_{↓}$, where $G_{↑}$ and $G_{↓}$ represent spin-up conductance and spin-down conductance, respectively. The condition $U/E_{F}$ is required to control the spin transport by varying the gate voltage [24]. In the case where the Dirac material is ferromagnetic graphene with a wave vector with $k_{F}L$ equal to 1 and a Fermi energy $E_{F}$ equal to 1, it can be shown that the length of the barrier is approximately 0.66 μm. The gate voltage and the exchange field require more than 1 meV for controllable spin transport. This structure can be reduced to a single tilted barrier by neglecting the width of the gap, and
(11)$$q_{x}^{\tau ,\pm }=p_{x}^{\tau ,\pm }=\frac{1}{L}\left(A\mp B\right)=\frac{1}{L}\left(\frac{\xi_{x}\omega }{1-{\xi_{x}}^{2}}\mp \frac{\sqrt{\omega^{2}+{\left(k_{F}L\sin\phi \right)}^{2}\left(1-{\xi_{x}}^{2}\right)}}{1-{\xi_{x}}^{2}}\right)$$

The new spin and tilt dependence transmission coefficient can be written as
(12)$$t_{\tau ,\xi }=\frac{e^{2iA}e^{-2ik_{F}L\cos\phi }\cos^{2}\phi }{\sin^{2}B\sin^{2}\phi +{\left(\cos B\cos\phi +i\sin B\right)}^{2}}$$

If the tilted velocity ratio is neglected, the transmission coefficient in Equation (12) can be reduced to that in [24].

## 3. Results and Discussion

In this study, we focus on the oscillatory characteristic of spin-resolved conductance in the presence of a TTDC within asymmetric tilted velocity barriers. The TTDC is characterized by a tilted velocity $\rho_{y1,2}$, as depicted in Figure 1a. Our investigation begins with the analysis of spin-resolved conductance oscillation behavior in the context of an asymmetric TTDC. The results, as shown in Figure 3, reveal an oscillatory behavior in spin-resolved conductance, $G_{↑}$ and $G_{↓}$. The presence of an asymmetric TTDC coupled with an exchange field results in a phase shift in the oscillation behaviors of spin-resolved conductance.

Interestingly, the spin conductance oscillates with a combination of two distinct patterns within a single period. The first half of the period shows a linear pattern, controllable by the gate voltage. This linear pattern arises from the superposition of both types of spin-resolved conductance, each oscillating with distinct phases. The second half of the period, on the other hand, exhibits an oscillatory pattern.

Comparing these findings with the results of spin conductance without the tilted parameter from [22], it is evident that the TTDC with an exchange field introduces the oscillating pattern in the second half period of spin conductance. When the difference in the tilted velocity ratio between two barriers ($\Delta \xi_{y}=\xi_{y2}-\xi_{y1}$) is equal to 0.4, the oscillatory pattern in the second half period for the spin conductance is analogous to the quantized conductance. However, no quantum beat pattern was observed in any conductance for the asymmetric tilted velocity barrier in the case of the TTDC.

Furthermore, we explore the influence of the TTDC on the oscillatory behavior of spin-resolved conductance by simplifying the model to a single TTDC. The results, as shown in Figure 4a, indicate that spin-up and spin-down components exhibit distinct phases of oscillation in spin-resolved conductance. Upon introducing varying values of the transverse-tilted velocity ratio, we observed oscillations in the conductance as a function of the gate voltage. However, it is noteworthy that the frequency of the spin conductance oscillation remains unaffected by the TTDC, indicating no discernible relationship between the tilted velocity and the frequency of conductance oscillation. We further observed that an increase in tilting of the TTDC leads to a splitting of the peaks in the oscillation pattern. This splitting is attributed to a shift in the electron’s wave vector, which is induced by the breaking of time reversal symmetry in the presence of a pseudo-magnetic field. A pseudo-magnetic field can be induced through two methods in materials with a tilted Dirac cone. The first occurs in a homogenous-tilted Dirac cone structure coupled with an electric field [17]. The second method involves the presence of an asymmetric-tilted Dirac cone junction, even without an applied electric field [44].

The presence of the TTDC leads to asymmetric refraction within the barrier, as depicted in Figure 4b. When the electrons are transported to the TTDC barrier, the wave vector with a positive incident angle bends towards the normal, while the wave vector with a negative incident angle bends away from the normal. From this scenario, the spin-resolved conductance with two regions of incident angles can be represented as follows:(13)$$G_{\tau ,\xi_{y}}=G_{\tau ,\xi_{y}}\left(\phi^{+}\right)+G_{\tau ,\xi_{y}}\left(\phi^{-}\right)=G_{0}\left(\int_{0}^{\pi /2}T_{\tau ,\xi }\left(\phi \right)\cos\phi d\phi +\int_{-\pi /2}^{0}T_{\tau ,\xi }\left(\phi \right)\cos\phi d\phi \right)$$
where $G_{\tau ,\xi_{y}}\left(\phi^{+}\right)$ and $G_{\tau ,\xi_{y}}\left(\phi^{-}\right)$ correspond to the contribution from the wave vector with positive and negative incident angles, respectively. The results, depicted in Figure 4c, confirm that the peak splitting observed in the spin-resolved conductance originates from the superposition of asymmetric refraction within the barrier. Both spin-resolved conductances exhibit the same asymmetric refraction behavior. For spin-up conductance, there is a higher probability of electron transmission with a wave vector possessing a positive incident angle compared to the wave vector with a negative incident angle. Conversely, for spin-down conductance, the chance of electron transmission with a wave vector having a negative incident angle is higher than that with a positive incident angle. In summary, our findings unveil the occurrence of anomalous spin conductance oscillation in the presence of an asymmetric TTDC combined with the exchange field. This observation offers a promising approach for identifying and characterizing the TTDC in spintronic devices.

Next, we turned our attention to exploring the behavior of spin-resolved conductance oscillation in the presence of an asymmetric LTDC system. In this configuration, the Dirac cone is tilted parallel to the direction of electron propagation. The asymmetry in the LTDC introduces unique properties that can affect the spin transport characteristics. To gain insights into the oscillation behavior in the case of an asymmetric LTDC Dirac cone, we initiated our investigation by examining the effect of tilted velocity in a single LTDC. By analyzing the spin-resolved conductance, we observed oscillatory behavior in both the spin-up and spin-down components. Notably, the phases of these oscillations differ between the two spin orientations. Furthermore, intriguingly, we discovered that the presence of the LTDC influences the oscillation frequency of the conductance, as shown in Figure 5. 

In this section, we delve further into the influence of a mismatched LTDC within an asymmetric tilted Dirac cone structure and its impact on the oscillatory behavior of spin- resolved conductance. We specifically investigated the effect of the LTDC mismatch on the oscillation patterns exhibited by spin-resolved conductance, focusing our analysis on the relationship between conductance and gate voltage for spin-up conductance. The decision to focus on spin-up conductance was motivated by the observation that the oscillatory behavior of spin-resolved conductance is highly similar. By narrowing our investigation to spin-up conductance, we can effectively capture the key features of the oscillatory patterns.

In Figure 6b, a distinct quantum beating pattern is depicted in the spin-resolved conductance, revealing an intriguing phenomenon. This behavior emerges from the superposition of two oscillating conductances within the asymmetric LTDC, as shown in Figure 6a. These spin-resolved quantum beats offer valuable insights into the complex dynamics of spin-resolved conductance, particularly in the presence of an asymmetric mismatched LTDC. To achieve a comprehensive understanding of the oscillating characteristic of spin-resolved quantum beats in the presence of the asymmetric mismatch LTDC, the oscillations in spin-resolved conductance can be precisely described by a cosine function.
(14)$$G_{\tau }\left(\xi_{x1}\right)\propto \cos^{2}\left(\frac{1}{1-{\xi_{x1}}^{2}}\left(\chi +\Lambda_{\tau }\mu \right)\right)\;\mathrm{and}\;G_{\tau }\left(\xi_{x2}\right)\propto \cos^{2}\left(\frac{1}{1-{\xi_{x2}}^{2}}\left(\chi +\Lambda_{\tau }\mu \right)\right)$$

To accurately determine the frequencies of beats in each spin-resolved conductance within a single barrier, we analyzed the transmission probability derived from the transmission coefficient, as described by Equation (9). In the limit of $E_{F}\ll \left|U+\Lambda_{\tau }J\right|$, the transmission probability for the single barrier can be expressed as follows:(15)$$T_{\tau }\left(\xi_{x1,2}\right)\to \frac{\cos^{2}\phi }{1-\sin^{2}\phi \cos^{2}\left(\frac{1}{1-{\xi_{x1,2}}^{2}}\left(\chi +\Lambda_{\tau }\mu \right)\right)}$$

In the case of a single barrier, we found that the beat frequency of the spin-resolved conductance is $f\approx \frac{1}{2\pi }\left(\frac{1}{1-{\xi_{x1,2}}^{2}}\right)$. The frequencies associated with each spin-resolved conductance exhibit a proportional relationship, as described by Equation (15). This proportionality provides a fundamental understanding of the observed oscillatory behavior in spin transport through a tilted Dirac cone.

Figure 6a,b provide a visual representation that effectively demonstrates the distinction between the two oscillatory components of spin-resolved conductance within single tilted Dirac cones. It illustrates how these components combine coherently, resulting in the observed conductance in double tilted velocity barriers. The figure uses a box with dashed lines to represent constructive interference and a box with solid lines to represent destructive interference, offering a clear visual depiction of these phenomena. Figure 6c combines the two sources for asymmetric double tilted velocity barriers. This equation provides a mathematical representation of how the two sources interact and contribute to the overall behavior observed in the system. The accompanying figure visually depicts this combination, enhancing our understanding of the complex dynamics at play in asymmetric double tilted velocity barriers. This visual representation enhances the understanding of the complex behavior and dynamics of spin transport in the appearance of tilted Dirac cones. The behavior of the superposition is characterized by the following expressions, which reveal the oscillatory nature of the spin conductance in the double tilted Dirac cone system:(16)$$G_{\tau }\left(\xi_{x1},\xi_{x2}\right)\propto \cos^{2}\left(\frac{1}{1-{\xi_{x1}}^{2}}\left(\chi +\Lambda_{\tau }\mu \right)\right)+\cos^{2}\left(\frac{1}{1-{\xi_{x2}}^{2}}\left(\chi +\Lambda_{\tau }\mu \right)\right)$$

The equation can be rearranged as follows:(17)$$G_{\tau }\left(\xi_{x1},\xi_{x2}\right)\propto \cos\left(\frac{1}{{\xi_{x}}^{+}}\left(\chi +\Lambda_{\tau }\mu \right)\right)\cos\left(\frac{1}{{\xi_{x}}^{-}}\left(\chi +\Lambda_{\tau }\mu \right)\right)$$
where $\frac{1}{{\xi_{x}}^{+}}=\frac{1}{1-{\xi_{x1}}^{2}}+\frac{1}{1-{\xi_{x2}}^{2}}$ and $\frac{1}{{\xi_{x}}^{-}}=\frac{1}{1-{\xi_{x1}}^{2}}-\frac{1}{1-{\xi_{x2}}^{2}}$. This result reveals the beat frequency term $\cos\left(\frac{1}{{\xi_{x}}^{-}}\left(\chi +\Lambda_{\tau }\mu \right)\right)$ and amplitude term $\cos\left(\frac{1}{{\xi_{x}}^{+}}\left(\chi +\Lambda_{\tau }\mu \right)\right)$. The number of quantum beats (loop) in the spin-resolved conductance is related to the difference in tilted velocity ratios between the two regions. As the relation between the spin-resolved conductance and the gate voltage, the beat frequency can be determined as $f_{b}={\xi_{x}}^{-}/2\pi$. Furthermore, it was shown that the presence of the exchange field leads to a shift in the spin-resolved conductance. To simplify the observation of spin-resolved beating behavior, an indirect observation of the charge conductance can be employed. This approach allows us to detect and analyze the oscillatory patterns in the conductance, which provides valuable insights into the spin transport properties in the system. The quantum beating in charge conductance oscillation is depicted in Figure 7. The oscillatory characteristic of the charge conductance arises from the superposition of both spin-resolved conductances as $G_{C}=G_{↑}\left(\xi_{x1},\xi_{x2}\right)+G_{↓}\left(\xi_{x1},\xi_{x2}\right)$. In the presence of an exchange field, both spin-resolved conductances undergo a phase shift. The phase shift between the two spin-resolved conductances is represented by 2μ. It is important to note that the phase shift for each individual spin-resolved conductance remains constant and is not influenced by the difference in the tilted velocity ratio between the two regions. This significant finding is visually presented in Figure 7a,b, where the results clearly illustrate the observed phase shift behavior. These figures provide a visual representation of the phase differences between the spin-resolved conductances, emphasizing the impact of the exchange field on the conductance oscillation. The consistent phase shift for each spin orientation underscores the robust nature of this phenomenon.

## 4. Conclusions

This study investigated the influence of tilted Dirac cones (TDCs) on spin transport properties within an ND/TDC1/ND/TDC2/ND junction configuration. We analyzed both transverse-tilted Dirac cones (TTDCs) and longitudinal-tilted Dirac cones (LTDCs) to understand their distinct effects. We observed oscillatory behavior in spin-resolved conductance. Notably, the transverse tilt does not induce quantum beating but leads to peak splitting in these oscillations. Second, we examined the LTDC in the ND/LTDC1/ND/LTDC2/ND junction. The presence of asymmetric tilts in LTDCs results in a unique quantum beating pattern. This effect arises due to the superposition of conductance oscillations with different frequencies, determined by the difference in longitudinal-tilt velocities between the two barriers. The phase shift for spin-resolved conductance remains at 2μ and is independent of the difference in tilted velocity from two barriers. These findings provide valuable insights into how the orientation and the degree of asymmetry of the tilt in a Dirac cone influences spin transport in two-dimensional materials. Understanding these effects is crucial for the development of spintronic devices and opening avenues for future research in spin manipulation techniques.

## Figures and Tables

**Figure 1 nanomaterials-14-00811-f001:**
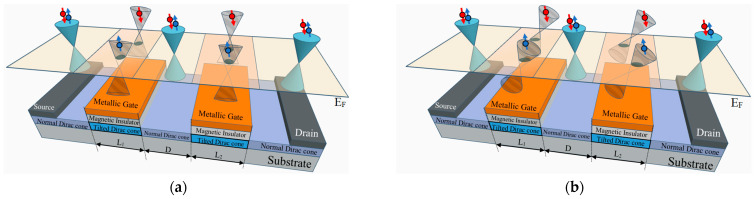
An asymmetrical double-tilted velocity barrier structure along with its corresponding Dirac cones. Regions with lengths $L_{1}$ and $L_{2}$ represent the asymmetry in the double-tilted velocity barriers, which are coated by a metallic gate on a magnetic insulator. The tilted Dirac cone within the double-tilted velocity barriers is depicted in two configurations: (**a**) perpendicular and (**b**) parallel to the direction of electron transport.

**Figure 3 nanomaterials-14-00811-f003:**
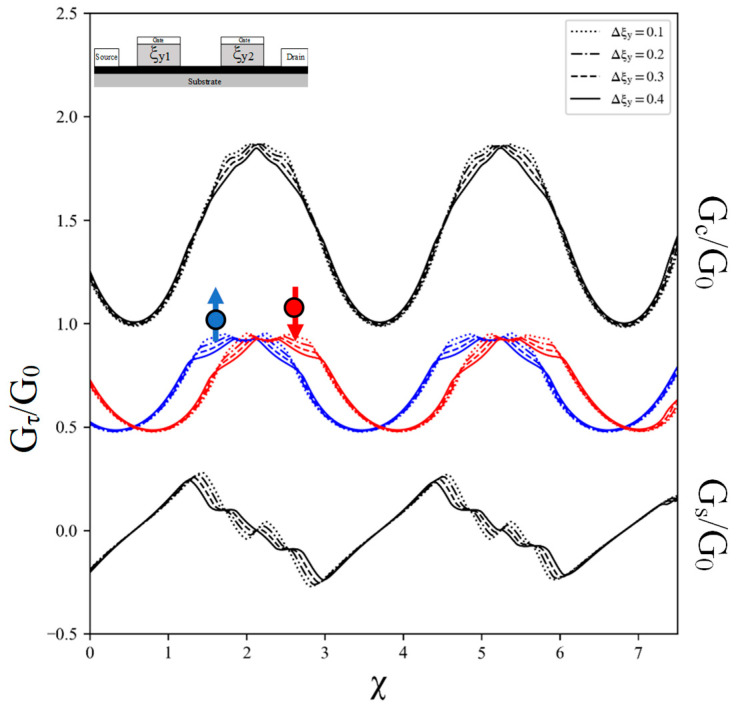
Plots of conductance versus gate voltage for charge conductance *G_C_*, spin-up conductance (blue), spin-down conductance (red), and spin conductance *G_S_* for double TTDC with different values of difference tilted velocity from two barriers $\Delta \xi_{y}=\xi_{y2}-\xi_{y1}$ when $\xi_{y1}$ is equal to 0.1. The Fermi energy $k_{F}L$ and the exchange field μ are fixed to 1 and 10, respectively.

**Figure 4 nanomaterials-14-00811-f004:**
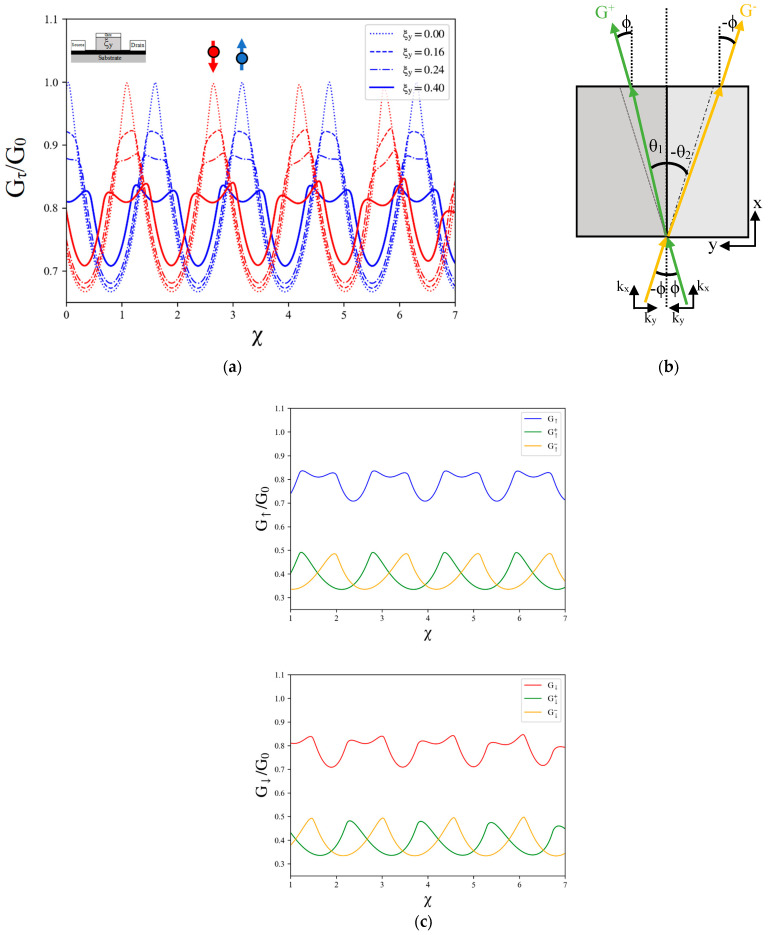
(**a**) Plots of conductance versus gate voltage for spin-up conductance (blue) and spin-down conductance (red) for single TTDC with varying values of transverse-tilted velocity ratio. The Fermi energy $k_{F}L$ and the exchange field μ are fixed to 1 and 10, respectively. (**b**) Asymmetric refraction of electron transport with different incident angles through a single tilted velocity barrier where the symmetry in the transmission is restored. (**c**) Plots of spin-up (blue) and spin-down (red) conductance versus gate voltage and the spin-resolved conductance with the incident region $G_{\tau ,\xi_{y}}\left(\phi^{+}\right)$ (green) and $G_{\tau ,\xi_{y}}\left(\phi^{-}\right)$ (yellow).

**Figure 5 nanomaterials-14-00811-f005:**
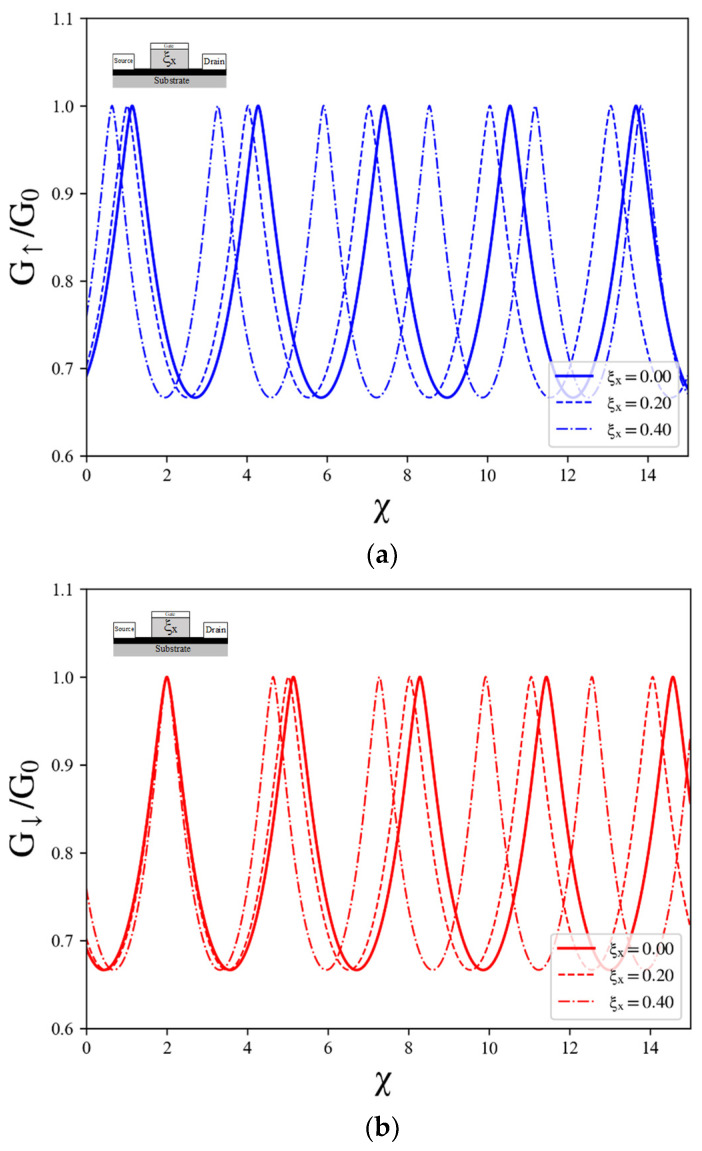
Plots of conductance versus gate voltage for (**a**) spin-up conductance (blue) and (**b**) spin- down conductance (red) in single tilted velocity barrier for LTDC with different values of tilted velocity ratio. The Fermi energy $k_{F}L$ and the exchange field μ are fixed to 2.

**Figure 6 nanomaterials-14-00811-f006:**
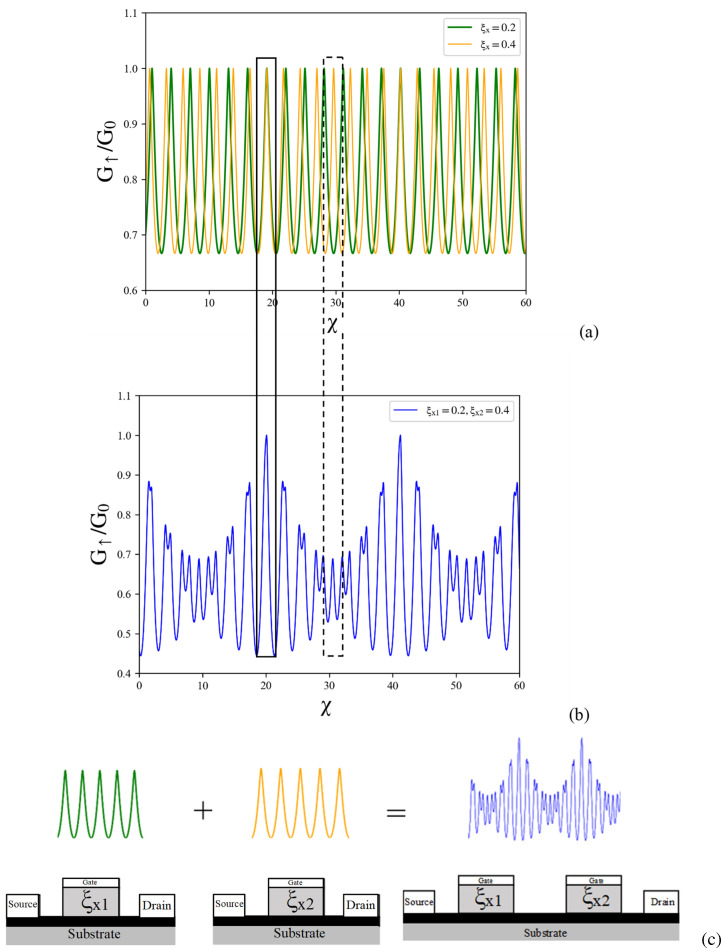
Plots of conductance versus gate voltage for (**a**) the spin-up conductance of electron in single longitudinal-tilted velocity barrier. (**b**) Spin-up conductance in double longitudinal-tilted velocity barriers and (**c**) the combination of two oscillatory components from the asymmetric longitudinal-tilted velocity barriers. The other parameters are given by $\xi_{x1}$ = 0.2 and $\xi_{x2}$ = 0.4, respectively.

**Figure 7 nanomaterials-14-00811-f007:**
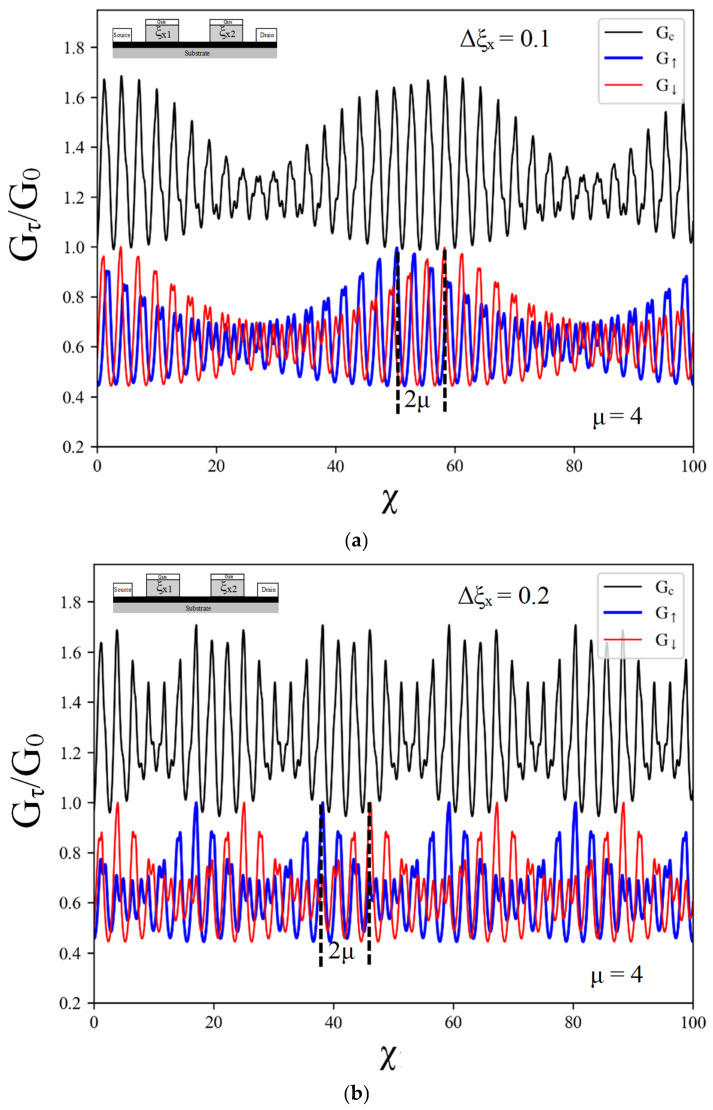
The charge conductance (black line) and spin-resolved conductance as a function of gate voltage for the spin-up conductance (blue line) and spin-down conductance (red line) in the asymmetric double longitudinal-tilted velocity barrier structure. The values of difference in tilted velocity from two barriers $\Delta \xi_{x}=\xi_{x2}-\xi_{x1}$ are (**a**) $\Delta \xi_{x}$ = 0.1 and (**b**) $\Delta \xi_{x}$ = 0.2 when $\xi_{x1}$ is equal to 0.2. The Fermi energy $k_{F}L$ and the exchange field μ are fixed to 2 and 4, respectively.

## Data Availability

Data are contained within the article.

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
