# Peer review of "Asymmetric Tilt-Induced Quantum Beating of Conductance Oscillation in Magnetically Modulated Dirac Matter Systems"

_nanomaterials, 2024, doi:10.3390/nano14090811_

Round 1
Reviewer 1 Report (Previous Reviewer 1)
Comments and Suggestions for Authors
Judging from the revised content, the author's understanding of the content and significance of his work has deepened, which is a good phenomenon. I hope the author can bring rigorous and scientific work to readers in the future.
Author Response
Thank you very much for your positive feedback on the revised manuscript. I appreciate you acknowledging the deepened understanding of the content and significance of the work. Your comments and suggestions have been invaluable in improving the quality of the manuscript.
Reviewer 2 Report (New Reviewer)
Comments and Suggestions for Authors
The paper examines the effect of tilt mismatch on the quantum oscillations of spin transport properties in two-dimensional asymmetrically tilted Dirac cone systems. The study deals with conductance oscillation in two distinct junction types: transverse and longitudinal tilted Dirac cones (TTDC and LTDC). First, findings reveal an unusual quantum oscillation of spin polarized conductance within the TTDC system, characterized by two distinct anomaly patterns within a single period, labeled as the linear conductance phase and the oscillatory conductance phase. These phases emerge in association with tilt-induced orbital pseudo-magnetization and exchange interaction. Next, they demonstrated that the structure of the LTDC can modify the frequency of spin conductance oscillation, and the asymmetric effect results in a quantum beating pattern in oscillatory conductance. An enhancement in the asymmetric longitudinal tilt velocity ratio within the structure correspondingly amplifies the beating frequency. Conclusions are as follows.
They study the influence of tilted Dirac cones (TDCs) on spin transport properties within a ND / TDC1 / ND / TDC2 / ND junction configuration. They analyze both transverse tilted Dirac cones (TTDC) and longitudinal tilted Dirac cones (LTDC) to understand their distinct effects. For TTDC, the transverse tilt does not induce quantum beating but leads to peak splitting in these oscillations. For LTDC in the ND / LTDC1 / ND / LTDC2 / ND junction, the presence of asymmetric tilt results in a unique quantum beating pattern. This effect arises due to the superposition of conductance oscillations with different frequencies, determined by the difference in longitudinal tilt velocities between the two barriers.
The results of the paper seem to be interesting. However, the paper is not suitable for the publication in the present form. The revision is recommended by taking account of the following comments.
1) There are many typos in several Equations, which prevent the correct understanding. For examples, parenthesis are missing on lines 220, 223, 224 and Eqs. (6),(8),(11),(15), and (16). Especially Eqs. (15) are duplicated.
2)The main result of the paper is the quantum beating pattern in the conductive oscillation of LTDC. When the direction of the tilted Dirac cone deviates from the longitudinal one as shown in Fig. 2, does such a beating still exist? It is useful to mention the range of tilting angle in line 221 for finding the beating.
Author Response
Please see the attachment.

Reviewer 3 Report (New Reviewer)
Comments and Suggestions for Authors
This manuscript reports on a theoretical investigation of spin transport in two-dimensional asymmetrically tilted Dirac cone systems. The authors study the oscillations of the conductance between two junctions involving transverse or longitudinally tilted Dirac cones. Based on their findings the authors argue that the quantum beating pattern in these conductance oscillations could be used to characterize Dirac fermions in type-I Dirac semimetal-based devices.
The topic of research is relevant for ongoing studies and exploitation of Dirac materials and would be suitable for a journal such as Nanomaterials. The theoretical investigation is sensibly structured and overall resonably well-presented. My only concern is with the numerical values for the parameters used by the authors in their calculations. It would be helpful to a reader interested in reproducing these results to clearly specify these for the different figures shown (or quantitatively argue why some are neglected). If the authors can satisfactorily address this issue this work might be publishable in Nanomaterials.
Comments on the Quality of English Language
The English is overall fine.
Round 2
Reviewer 3 Report (New Reviewer)
Comments and Suggestions for Authors
The authors have satisfactorily addressed my comments and I can give a positive recommendation for publication in Nanomaterials.
Comments on the Quality of English Language
The English is fine.
This manuscript is a resubmission of an earlier submission. The following is a list of the peer review reports and author responses from that submission.
Round 1
Reviewer 1 Report
Comments and Suggestions for Authors
The authors report oscillatory behavior in spin-resolved conductance with the ferromagnetic Dirac material ND / TDE1 / ND / TDE2 / ND junction. This work looks good. The content of the theoretical work itself is already very detailed. Therefore, the authors should provide specific material structure possibilities based on the predicted structure of their work, and therefore discuss in detail whether their theoretical work has experimental significance. During this process, please refer to existing published work by others as much as possible. Please provide specific examples for the experimental structure, so that more scientists conducting experiments can understand the significance of their work.
Reviewer 2 Report
Comments and Suggestions for Authors
Dear Editor,
As this manuscript stands at the moment, it is simply too difficult to read to even try to evaluate it. This difficulty does not stem from the subject, but rather from an incomprehensible presentation, misuse of the English language, incomplete statements, incoherent captions, undefined acronyms, and an arbitrary starting point. While some parts of the model might have been introduced elsewhere, the Authors should make an effort to guide the reader of this manuscript through the relevant steps starting from a common solid background.
Therefore my suggestion is that the manuscript be rewritten in each and any of its parts. I must add that the introduction reads somewhat coherently, although even there there are a couple of hung statements.
My recommendation is, thereby, rejection.
Comments on the Quality of English Language
The language is misused, often wrong, with incomplete or hung statements that impede the reading of the manuscript.